# Short-Term Dietary Intervention with Cooked but Not Raw *Brassica* Leafy Vegetables Increases Telomerase Activity in CD8+ Lymphocytes in a Randomized Human Trial

**DOI:** 10.3390/nu11040786

**Published:** 2019-04-05

**Authors:** Hoai Thi Thu Tran, Nina Schlotz, Monika Schreiner, Evelyn Lamy

**Affiliations:** 1Molecular Preventive Medicine, IUK, University Medical Center and Faculty of Medicine, University of Freiburg, 79106 Freiburg, Germany; hoai.tran@uniklinik-freiburg.de (H.T.T.T.); nina.schlotz@uni-konstanz.de (N.S.); 2Pharmaceutical Bioinformatics, Institute of Pharmaceutical Sciences, Albert-Ludwigs-University, 79104 Freiburg, Germany; 3Department of Biology, University of Konstanz, 78457 Konstanz, Germany; 4Leibniz Institute of Vegetable and Ornamental Crops, 14979 Großbeeren, Germany; schreiner@igzev.de

**Keywords:** food intervention, *Brassica*, polyphenols, isothiocyanates, telomerase activity, T lymphocytes

## Abstract

Telomerase in T lymphocytes is dynamic and limited evidence from epidemiological studies indicates that the enzyme can be modulated in peripheral lymphocytes by dietary and lifestyle factors. The differential effect of dietary intervention on T cell subsets has not been investigated so far. *Brassica* vegetables are known for their multiple beneficial effects on human health, and here, the effect of a five-day short-term intervention with raw or cooked leaves of *Brassica carinata* on telomerase activity in CD4+ and CD8+ T cells from 22 healthy volunteers was investigated in a randomized single-blind, controlled crossover study. Blood samples were collected before and after intervention, and CD4+/CD8+ T lymphocytes were isolated. Telomerase activity was quantified using the TRAP-ELISA assay. Intervention with both preparations led to a marginal increase in telomerase activity of CD4+ cells compared to the baseline level. In CD8+ cells, a significant increase in telomerase activity (25%, *p* < 0.05) was seen after intervention with the cooked material. An increase in telomerase activity in CD8+ cells of healthy volunteers could be regarded as beneficial in terms of helping with the cell-mediated immune response. Whether a *Brassica* intervention has long-term effects on telomere extension in specific T cell subsets needs to be determined.

## 1. Introduction

Telomerase is a ribonucleoprotein enzyme with reverse transcriptase activity that is responsible for extension of telomeric DNA [1]. In humans, expression of telomerase is suppressed in most normal somatic cells, whereas it is upregulated in lymphocytes during certain stages of development, differentiation, and upon activation. This has been found to confer expanded replicative capacity on these cells. With the loss of telomerase activity, T lymphocytes undergo progressive telomere shortening as they continue to divide [2]. Importantly, telomerase activity upregulation was shown to enhance humoral and cell-mediated immune responses [3]. Low telomerase activity is associated with an increased risk of non-communicable diseases (NCD) including diabetes, coronary heart disease (CHD) and cancer [4,5]. Few studies have demonstrated that dietary or lifestyle intervention can increase telomerase activity in human peripheral mononuclear cells (PBMC) [6,7]. However, quantification of total PBMC telomerase reflects a mixture effect which might be difficult to interpret as it does not offer any mechanistic explanation about what types of immune cells are driving the findings. Since PBMC are composed of many cell types and the percentage of each cell type varies from individual to individual, it is also difficult to discern the role of telomerase regulation in mediating the effects of food intervention.

Ethiopian kale (*Brassica carinata)* is a plant species of the botanical family *Brassicaceae*. *Brassicaceae* have been particularly investigated for their potential of interference with the development of NCD [8,9,10,11,12,13]. Their leaves are a rich source of nutrients with high concentrations of bioactive phytochemicals, especially polyphenols and the glucosinolate-myrosinase degradation products isothiocyanates (ITC) [14,15,16]. The reported health effects include anti-inflammatory, immunomodulatory, and cancer preventive properties [17,18]. The vegetables are mainly eaten cooked, not raw, but processing might affect the health-promoting potential of the plant as suggested earlier [19]. Thus blood samples from a short-term intervention trial with two different consumption forms (raw versus cooked) of *B. carinata* leaves were used to evaluate the impact on telomerase activity in CD4+ and CD8+ T cell subsets from healthy human volunteers.

## 2. Materials and Methods

### 2.1. Chemicals

Phosphate buffered saline (PBS, without Ca and Mg) was purchased from Gibco™, Life Technologies GmbH (Darmstadt, Germany). Quick Start Bradford 1× Dye Reagent was purchased from BioRad Laboratories GmbH (Munich, Germany). Trypan blue was purchased from Sigma-Aldrich Chemie GmbH (Taufkirchen, Germany). Anti-CD3 (clone REA613) mAb, anti-CD4 (clone 15E8) mAb, and anti-CD8 (clone BW135/80) mAb were from Miltenyi Biotec (Bergisch Gladbach, Germany).

### 2.2. Subjects and Study Protocol

This study was carried out on blood samples derived from a single-blind randomized controlled crossover intervention study at the University Medical Center in Freiburg, Germany as reported earlier [20]. This study was conducted according to the Declaration of Helsinki and all procedures involving human subjects were approved by the Ethics Committee of the University of Freiburg (ethical vote number 277/16). The study design is given in Figure 1.

Recruitment started on 5 September 2016 and the last intervention phase ended on 18 November 2016. Details of the design are also given in the German Clinical Trials Register (DRKS00010836, 26 July 2016). Sample size calculation was done as reported in Schlotz et al. [20]. Assuming a drop-out rate of 20%, 24 subjects were enrolled in the original trial since 20 subjects were required for detecting an effect of 25% difference in the primary parameter (DNA damage), in the response to matched pairs with 20% standard deviation at 90% power and α = 0.01 significance. Quantification of telomerase activity was a secondary parameter in the reported study. Frozen cell samples were from all 22 healthy young adults (5 males and 17 females, aged 20–25 years) with a body mass index (BMI) of >18 and <26, and were non-smokers. We asked the participants to avoid intense physical activity during the intervention phases. Written informed consent was provided by all subjects and health questionnaires were used to qualify their eligibility. Random allocation was done using Microsoft^®^ Excel. Each subject attended two intervention phases, both intervention phases were preceded by a seven-day wash-out phase consisting of a special glucosinolate/ITC-free and polyphenol-reduced diet. The special diet continued for the time of the intervention phase. More details are given in Schlotz et al. [20]. During intervention, the participants consumed one dose of 15 g freeze-dried powder of *B. carinata* leaves, freshly mixed with 200 mL of water every morning for five days under the supervision of the study coordination. The given dose of freeze-dried powder equaled to 150 g fresh leaves of *B. carinata*. The sequence of consumption (start with raw or cooked plant preparation) was allocated randomly.

### 2.3. Blood Collection and Isolation of CD4+ or CD8+ T cells

Blood was taken in the morning between 8:00 to 11:00 a.m. On day one, this was before intervention (baseline), and on day five this was two hours after the fifth dosing (intervention) in Li-Heparin vacutainers. Within 2 h, CD4+ or CD8+ T lymphocytes were isolated from blood using the EasySep™ Direct Human CD4+/CD8+ T Cell Isolation Kit (StemCell Technologies, Cologne, Germany) according to the manufacturer’s instructions. Isolated CD4+ and CD8+ T lymphocytes from 5 mL fresh blood were pelleted and stored at −80 °C until further processing.

### 2.4. Phytochemical Analysis in the Plant Material

The preparation of the freeze-dried powder of *B. carinata* leaves material is already described elsewhere [15]. One preparation contained either freeze-dried powder from raw leaves of *B. carinata*, or from shortly cooked (10 min, 200 g per 4 L water) *B. carinata* leaves. Analysis of the main bioactive phytochemicals, namely allyl ITC and polyphenols, showed that 15 g of the plant powder preparation freshly mixed with 200 mL tab water contained either 177 ± 3.02 µmol (*n* = 3) allyl ITC (raw) or no allyl ITC (cooked). The amount of total polyphenols was comparable in both preparations. The raw material contained a total of 108 ± 14.6 mg polyphenols and the cooked 97 ± 5.25 mg (*n* = 6). The details of the chemical analysis are given in Reference [20].

### 2.5. Telomerase Activity Analysis by TRAP-ELISA Assay

Telomerase activity was assessed by using the TeloTAGG Telomerase PCR ELISA Kit from Sigma-Aldrich Chemie GmbH (Taufkirchen, Germany) as described before [21]. Briefly, isolated CD4+/CD8+ T cells were lysed in lysis buffer (10,000 cells/µL) for 30 min, at 4 °C, and then centrifuged at 16,000× *g*, 20 min at 4 °C. The protein concentration was determined in the supernatant according to the method of Bradford [22]. For the PCR reaction, a total volume of 50 µL with equal amounts of protein was used and the products were amplified using 30 cycles (94 °C for 30 s, 50 °C for 30 s and 72 °C for 90 s). Each sample was run in duplicate; as negative controls, heat-treated samples, nuclease-free water, and lysis solution were used. Five microliters of the PCR product were used for the ELISA assay. Absorbance was measured at 450 nm with a reference wavelength of 690 nm using a multiplate reader from Tecan (Tecan Group Ltd, Crailsheim, Germany).

### 2.6. Flow Cytometry

Isolated T cell subpopulations were stained for surface expression of cell markers using an anti-CD3-FITC mAb, together with an anti-CD4-PE mAb or anti-CD8-PE mAb. Subsequently, the cells were analyzed by flow cytometry using a FACSCalibur™ (BD Biosciences, Heidelberg, Germany).

### 2.7. Statistical Analysis

Data were analyzed using GraphPad Prism 6.0 software (La Jolla, CA, USA). The data followed a normal distribution as determined by D’Agostino–Pearson omnibus test and were analyzed for statistical significance using Student’s paired *t*-test. *p*-values < 0.05 (*) were considered statistically significant.

## 3. Results

Telomerase activity in CD4+ and CD8+ T cell subsets was determined in samples from 22 healthy individuals. The participants were 22.7 ± 2.4 years old, with an average body weight of 70 ± 10.6 kg, height of 1.7 ± 0.1 m, and with a BMI of 23 ± 1.9 kg/m2 (see also Schlotz et al. [20]). The purity of isolated CD4+ or CD8+ T cells was >90% as analyzed by flow cytometry using anti-CD3 mAb staining together with anti-CD4 or anti-CD8 mAbs, shown in Figure 2A. Cell viability was >95% in all samples as determined by the trypan blue exclusion assay (data not shown). The mean cell number in the isolated subsets was comparable between time points (baseline vs. cooked food intervention: 1.4 ± 0.7 × 10^6^ vs. 1.3 ± 0.7 × 10^6^ CD4+ cells/mL and 0.7 ± 0.4 × 10^6^ vs. 0.6 ± 0.4 × 10^6^ CD8+ cells/mL; baseline vs. raw food intervention: 1.3 ± 0.5 × 10^6^ vs. 1.2 ± 0.6 × 10^6^ CD4+ cells/mL and 0.7 ± 0.4 × 10^6^ vs. 0.6 ± 0.4 × 10^6^ CD8+ cells/mL). The results are also given in Figure 3B.

Next, a range was determined, in which detection of telomerase enzyme activity in CD4+ and CD8+ cell lysate was clear positive (i.e., absorbance of >0.2), linear (or more importantly not saturated) and in a sensitive detection range, i.e., absorbance of ≤2.0. The results are given in absolute absorbance values (AU) in Figure 2B.

Based on these results, the amount of 0.08 µg protein was applied for telomerase quantification in the samples from the intervention trial. In Figure 3, the effect of the food intervention on the level of telomerase in T cell subsets is given. At baseline, telomerase activity was 0.92 ± 0.26 AU and 0.87 ± 0.25 AU in CD4+ cells from volunteers in the cooked and raw *B. carinata* leaves intervention group, respectively (Figure 3A). Both types of intervention did not result in a statistically significant change in telomerase activity in CD4+ T cells (intervention with cooked leaf material: 1.06 ± 0.57 AU; intervention with raw leaf material: 1.07 ± 0.56 AU). For CD8+ cells, the baseline was at 0.81 ± 0.41 AU and 0.77 ± 0.35 AU in the cooked and raw *B. carinata* leaves intervention group, respectively (Figure 3B). After intervention, a significant telomerase upregulation of 25% compared to baseline level was observed in samples from the cooked (1.02 ± 0.44 AU), but not from raw (0.89 ± 0.42), *B. carinata* leaves intervention.

## 4. Discussion

In normal human T cells, telomerase activity is tightly regulated and plays an important role in immune function in terms of protecting cells from telomere shortening [3].

Thus defining the potential of specific foods to increase telomerase activity in healthy humans could support the development of intervention strategies for maintaining or even enhancing immune cell function. Well controlled longitudinal intervention studies with repeated telomerase measures within individuals are scarce. So far, it was shown that a 16-week vitamin D supplementation significantly increased telomerase activity (19%) in PBMC from 19 obese African Americans [6]. Further, a 12-week micronutrient supplementation also significantly increased telomerase activity of PBMC (>25%) in 66 healthy women which paralleled an increase in antioxidant capacity [23]. An association of higher adherence to a Mediterranean diet rich in vegetables and fruits with increased telomerase activity was shown in PBMC from 79 elderly people in comparison with lower adherence groups [24]. A three-month lifestyle intervention study with 24 low-risk prostate cancer patients including meditation, a plant-based diet, and moderate exercise resulted in a 30% increase in telomerase activity of PBMC. Then, in the five-year follow-up, a significant telomere length extension was shown in 10 participants [7,25]. All of these studies were conducted with PBMC, providing information on the average telomerase activity. However, it is not really clear what the average telomerase activity in PBMC truly represents, and this is currently a great limitation for the significance of the studies. In healthy subjects, isolated PBMC consist of lymphocyte subpopulations (50–75% T cells, 5–15% B cells, and 5–15% natural killer cells) and monocytes, in which monocytes represent 5–20% of the population. Besides T cells, B and natural killer cells can express telomerase enzyme to a different extent [26,27,28]. For T cell subpopulations, telomerase activity has been reported to be slightly lower in CD8+ cytotoxic cells than CD4+ helper cells [29] which could be confirmed in the present study. It gets even more complex, since the ratio of CD4+/CD8+ cells in healthy subjects is known to be in range of 1.2–2:1 [30,31]. Consequently, a major part of PBMC telomerase quantification might originate from the CD4+ subset. Without studying telomerase dynamics in different T cell subsets the data interpretation will be compromised, activity changes in subsets could only be poorly detected or not at all. In the present study, the cooked *B. carinata* leaves intervention resulted in a statistically significant increase in telomerase activity in CD8+ but not CD4+ T lymphocytes, although a trend for increased activity was evident also in the CD4+ subpopulation. The small sample size should be considered here as a possible limitation of the study. It has been shown that, e.g., in vitro exposure to cortisol also affects telomerase activity in the two T cell subsets in a differential way in which the enzyme inhibition was stronger in CD8+ than CD4+ cells isolated from the same subject [32].

Thus, this study provides, for the first time, evidence that telomerase activity in T cells can respond quickly (within less than a week) to a food intervention. With 25%, the enzyme upregulation is in the range of previous observations made with PBMC. Whether this regulation would also become evident when studying PBMC instead of the T cell subsets is unknown since this was not investigated in parallel here. Basically, higher endogenous telomerase activity has been associated with better immune function and health status [24,33]. The constant expression of telomerase at low level in normal PBMC has been suggested to exert a function in proliferation of naïve and/or memory cells to maintain T cell homeostasis [34]. So, elevated telomerase activity in CD8+ T cells might help to maintain a healthy cytotoxic lymphocyte function and enhance their anti-viral ability. Using hTERT-transduced CD8+ T cells, it has been suggested that maintenance of telomerase activity is capable to extend life span and preserve the cytotoxic properties of CD8+ cells [35]. However, whether this observation is translatable to the in vivo situation remains unanswered.

It is thought that an intervention study of more than four months is necessary to observe reliable changes in telomere length [36]. The present study was designed as short-term intervention without follow-up. Thus, it needs to be determined whether the observed effect on telomerase activity could really compensate for telomere shortening in the CD8+ T cell subset under long-term conditions. In general, the length of telomere repeats reflects the balance between additions and losses. Telomerase and recombination can elongate telomeres, but in most somatic cells, additions are outbalanced by losses [37]. CD8+CD28− T cells were shown to have the shortest telomeres and lowest telomerase activity of all cell subtypes suggesting that they are most vulnerable to dysfunctional telomere-induced cellular senescence [29].

In the present study, the mean baseline level of telomerase activity varied in CD4+ or CD8+ T cells only by 5% between the two intervention phases which were separated by a 16-day break. This suggests that the observed effect of food intervention on telomerase is only transient. 

Telomerase in lymphocytes is known to follow a circadian rhythm. It was reported to be high in the early morning hours and then gradually declines following time via CLOCK/BMAL1 protein heterodimers [38]. We sampled the blood from all participants within a 2 h time slot at early morning hours. Therefore, we can exclude that the higher telomerase level in CD8+ subset post-intervention is artificial due to endogenous circadian oscillation.

In the cooked plant material from *B. carinata*, allyl ITC was absent and thus it can be concluded that ITCs do not account for the detected increase in telomerase of CD8+ cells. According to the phytochemical analysis of *B. carinata* leaves, the total amount of polyphenolic compounds was comparable in the raw and cooked plant material [15,20,39] and some of the polyphenols, such as caffeoylquinic acid, have also previously reported to be highly thermostable [40]. Although several complex flavonoids were significantly degraded by the cooking process, less complex compounds, for instance, kaempferol glycosides, were enriched [15,20]. So far, for some polyphenols and ITC, telomerase modulation in terms of inhibition has been shown in human cancer cells. This is regarded a promising pharmacologic strategy to drive cancer cells into apoptosis [21,41,42,43] However, the applied concentrations are usually over an order of magnitude higher than the plasma levels reached by dietary intervention. No in vitro study has been carried out so far to investigate the impact of these phytochemicals at physiologic concentrations on telomerase in normal cells of the immune system, and thus it remains a speculation to which plant compound the effect could be attributed.

## 5. Conclusions

Our findings suggest that telomerase activity of T cell subsets may be regulated by *Brassica* vegetable consumption, but processing condition seems to play a decisive modulating factor here. However, this statement is limited by the small sample size. The immune system strongly depends on clonal expansion and cell division and, thus, has evolved telomerase regulation as a strategy for telomere maintenance in support of this need. In this context, the present findings could support the theory of a beneficial effect. But whether this modulation is really an indicator of improved health or rather a compensatory mechanism induced by adverse conditions still needs further clarification. The results are in line with other observations by our group demonstrating strong anti-genotoxic capacity [20] as well as DNA repair capacity [44] in volunteers’ PBMC upon cooked *B. carinata* intervention.

## Figures and Tables

**Figure 1 nutrients-11-00786-f001:**
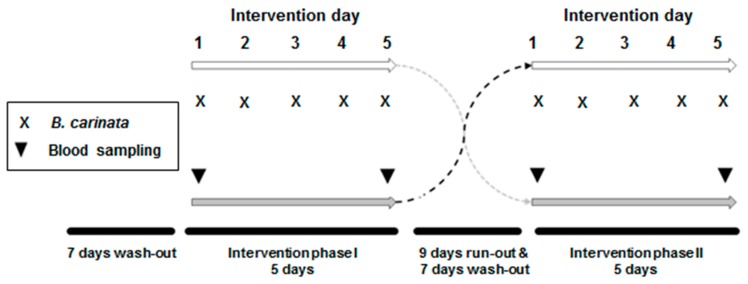
Design of the randomized crossover intervention trial.

**Figure 2 nutrients-11-00786-f002:**
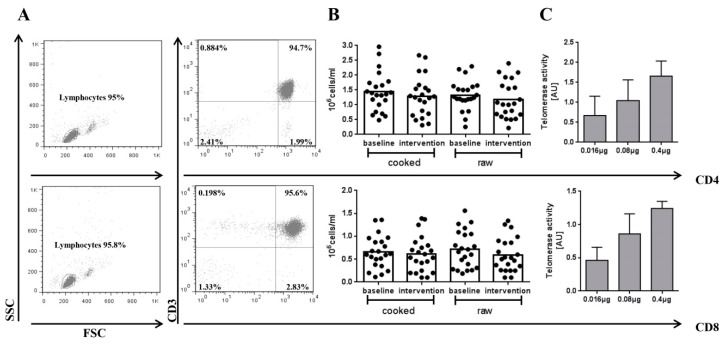
(**A**) Purity of CD4+ and CD8+ T cell subsets and titration of telomerase activity. A forward scatter/side scatter (FSC/SSC)-plot was made and all lymphocytes were gated. The lymphocyte population was copied to a CD3+/CD4+- or CD3+/CD8+-scatterplot identifying specific T-cell subsets. (**B**) Number of CD4+/CD8+ T cells in 1 mL of blood. (**C**) Titration of isolated CD4+/CD8+ T cells was done using the TRAP-ELISA assay. Bars are means + SD, *n* ≥ 3.

**Figure 3 nutrients-11-00786-f003:**
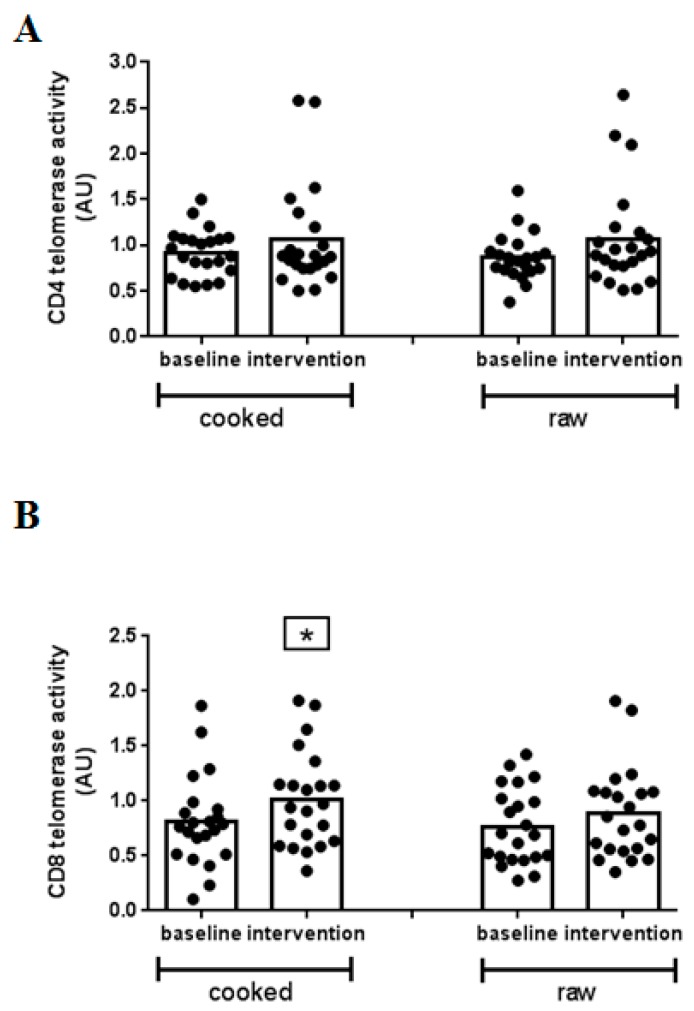
Effect of *B. carinata* intervention on telomerase activity in CD4+ and CD8+ T cell subsets. Telomerase activity was measured in isolated CD4+ T helper (**A**) and CD8+ cytotoxic T cells (**B**) at baseline and post intervention using the TRAP-ELISA assay. Each sample was measured in duplicate. Each dot represents one volunteer. The bars are mean values from *n* = 22 volunteers. Asterisks indicate statistically significant differences between baseline and intervention phase (*p* < 0.05).

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
