# Peer review of "Short-Term Dietary Intervention with Cooked but Not Raw Brassica Leafy Vegetables Increases Telomerase Activity in CD8+ Lymphocytes in a Randomized Human Trial"

_nutrients, 2019, doi:10.3390/nu11040786_

Round 1
Reviewer 1 Report
This study shows that short term cooked Brassical leafy vegetable intervention increased telomerase activity in CD8+ lymphocytes. The results are interesting. This is one of the few studies measuring the effect of intervention on telomerase activity in a specific type of lymphocytes.
I have a few comments:
In statistical analysis, please describe the distribution of telomerase activity, was it normally distributed? The dependent variable should be approximately normally distributed for the t-test.
Only cooked, not raw Brassical leafy vegetable increased telomerase activity, what compound could explain this difference? Did allyl ITC play a role in this differential finding? On the other hand, there was similar trend between cooked and raw in both CD4+ and CD8+ cells in Figure 3. Could the negative findings in raw or in CD4+ cells be due to small sample size and inadequate power?
The sample size is small, which should be acknowledged as a limitation. Please discuss the limitation of this study.
Author Response
In statistical analysis, please describe the distribution of telomerase activity, was it normally distributed? The dependent variable should be approximately normally distributed for the t-test.
We thank the reviewer for the comments. Normal distribution of telomerase activity data was tested using the D'Agostino-Pearson omnibus test, performed with the software Graphpad Prism 6. We now added this information in the M&M part.
Only cooked, not raw Brassica leafy vegetable increased telomerase activity, what compound could explain this difference? Did allyl ITC play a role in this differential finding?
On the other hand, there was similar trend between cooked and raw in both CD4+ and CD8+ cells in Figure 3. Could the negative findings in raw or in CD4+ cells be due to small sample size and inadequate power?
The sample size is small, which should be acknowledged as a limitation. Please discuss the limitation of this study.
As we described in our manuscript, allyl ITC was not present in the cooked material and thus we concluded that allyl ITC did not contribute to the sign. effect in CD8+ cells. However, we discussed the potential relevance of polyphenolic compounds (see p. 7, l. 235-247). The sample size was calculated for another parameter (DNA damage) as described in M&M (p.3, l. 77-81). Thus, it might be true that the insignificant telomerase activity increase in CD4+ cells could be due to a small sample size. We considered this in the conclusion part: “However, this statement is limited by the small sample size.”
Reviewer 2 Report
1. Is there any difference of T cells numbers before and after intervention?
2. The study design showed there are two intervention phases. It is not clear at which intervention phase that T cells were collected? In addition, the authors should compare T cells telomerase activity at first intervention phase, wash out phase, and the second intervention phase. This could be helpful to confirm the beneficial effects of cooked Brassica vegetables in inducing telomerase activity.
Author Response
1. Is there any difference of T cells numbers before and after intervention?
We thank the reviewer for this critical point. There was no change observed. We now added a figure showing these results in p.4, l. 137-141.
2. The study design showed there are two intervention phases. It is not clear at which intervention phase that T cells were collected? In addition, the authors should compare T cells telomerase activity at first intervention phase, wash out phase, and the second intervention phase. This could be helpful to confirm the beneficial effects of cooked Brassica vegetables in inducing telomerase activity.
We conducted an intervention trial with a crossover design, which means that every donor consumed both plant preparations (raw and cooked). Blood was taken before (day 1) and after 5x consumption (day 5) in both intervention phases. T cell subpopulations were directly isolated within 2 hrs from this blood. Consequently, we had 4 samples from each donor (before/after consumption of raw food and before/after consumption of cooked food). This design is described in p.2-3, l. 69-93. We also explained in the manuscript that telomerase activity at baseline level (before each intervention start) differed only marginally between the two phases in both CD4+ and CD8+ T cells. p.5, l. 154-163. Thus the observed effect can be attributed to the food intervention.
Reviewer 3 Report
I feel this manuscript is good, results are convincing and author justify them well. I would request to authors to read whole manuscript carefully, I found few typos mistakes and if possible to improve the graph scale.
Best Wishes
Author Response
We thank the reviewer for the positive comments. The manuscript was revised carefully with some modifications.
Reviewer 4 Report
The manuscript “Short-term dietary intervention with cooked but not raw Brassica leafy vegetables increases telomerase activity in CD8+ lymphocytes in a randomized human trial” reports the results of a study aimed at investigating the effect of raw vs cooked Etiopian kale leaves on telomerase activity in T cells.
The topic is interesting, however some details are missing and should be added in a revised version of the manuscript.
In detail, major comments are related to the design of the human intervention study.
Authors did not mention details about the diet consumed by the volunteers during the intervention phases. Did you collect any information about the diet? Did you provide a list of allowed and not allowed foods? Moreover, given that physical activity may influence telomerase length and activity, authors should clarify i) if high physical activity was an exclusion criteria and ii) if physical activity during the intervention phases was monitored
Authors should mention why the intervention study was just 5-day long. As properly mentioned in the discussion, most of the studies aimed at investigating the effect of specific diet components on telomerase activity last several weeks.
Lastly, authors should discuss in more depth why an effect was observed in CD8+ but not in CD4+ lymphocytes.
MINOR COMMENTS
Figure 2: the reference to figure 2A is missing. “(A)” probably should be added just before “Purity”
Line 106: how did you express the total phenolic content?
Author Response
Authors did not mention details about the diet consumed by the volunteers during the intervention phases. Did you collect any information about the diet? Did you provide a list of allowed and not allowed foods? Moreover, given that physical activity may influence telomerase length and activity, authors should clarify i) if high physical activity was an exclusion criteria and ii) if physical activity during the intervention phases was monitored.
Authors should mention why the intervention study was just 5-day long. As properly mentioned in the discussion, most of the studies aimed at investigating the effect of specific diet components on telomerase activity last several weeks.
We thank the reviewer for the feedback. The details regarding the special GLS/ITC-free and polyphenol-reduced diet during the intervention and wash out phase were given in a previous paper from our group (Schlotz, Odongo et al. 2018). We now added this more precisely in the methods section. Further, participants were asked to avoid intense physical activity during the study. We also added this information in the manuscript. p. 3, l. 83-84.
As correctly stated by the reviewer, so far published dietary intervention studies reporting on telomerase activity increase in PBMC were conducted for several weeks. However, it is known that telomerase activity is a dynamic factor that can quickly change. For example, in vitro mitogenic T cell stimulation or acute stress of volunteers has been shown to elevate telomerase levels in PBMC within hours or days (Epel, Lin et al. 2010); further, telomerase activity is known to change following a circadian rhythm (Chen, Wen et al. 2014). Thus, based on this knowledge we wanted to determine whether really long-term intervention with a diet rich in bioactive phytochemicals is necessary or if already a short-term intervention can lead to significant changes in the enzyme activity. Our findings now confirm that a short-term vegetable intervention can induce telomerase in T cell subsets in the range comparable to the one reported in earlier studies.
Lastly, authors should discuss in more depth why an effect was observed in CD8+ but not in CD4+ lymphocytes.
Based on the comments of reviewer one, we acknowledged the possibility that the insignificant telomerase activity increase in CD4+ cells could also be due to a small sample size and included this in the discussion part. There is only very limited information about the regulation of telomerase activity in T cell subsets. Thus, besides we further added in the discussion on that issue “…although a trend for increased activity was evident also in the CD4+ subpopulation. The small sample size should be considered here as a possible limitation of the study. It has been shown that e. g. in vitro exposure to cortisol also effects telomerase activity in the two T cell subsets in a differential way in which the enzyme inhibition was stronger in CD8+ than CD4+ cells isolated from the same subject (Choi, Fauce et al. 2008).”
MINOR COMMENTS
Figure 2: the reference to figure 2A is missing. “(A)” probably should be added just before “Purity”.
Figure 2 was modified.
Line 106: how did you express the total phenolic content?
The analysis of phenolic compounds was described in detail in our previous papers (Schlotz, Odongo et al. 2018) (Odongo, Schlotz et al. 2018). A total of 36 compounds were detected in the plant matrix. Phenolic compounds were analysed using HPLC (Agilent HPLC series 1100, Agilent Technologies Sales & Services GmbH & Co. KG, Waldbronn, Germany) coupled with an ion trap mass spectrometer (Bruker Amazon SL, Bruker, Bremen, Germany) and UHPLC (Agilent Technologies 1290 Infinity II,) coupled with quadrupole time of flight mass spectrometry; QToF-MS (Agilent Technologies 6230 TOF LC/MS-) equipped with an atmospheric pressure chemical ionization source (APCI) (Agilent Technologies) was used for the analyses of carotenoids and chlorophylls. Quantification of carotenoids and chlorophylls was done using external standard calibration curves of each compound. Quantification of phenolic compounds was done using caffeoylquinic acid (chlorogenic acid), quercetin 3-O-glucoside, and kaempferol 3-O-glucoside (Carl Roth, Karlsruhe, Germany) for external calibration curves in a semiquantitative approach. The identification of hydroxycinnamic acids and flavonoid glycosides was done as described by Neugart, Baldermann et al. (2017).
Cited literature:
Chen, W.-D., M.-S. Wen, S.-S. Shie, Y.-L. Lo, H.-T. Wo, C.-C. Wang, I. C. Hsieh, T.-H. Lee and C.-Y. Wang (2014). "The circadian rhythm controls telomeres and telomerase activity." Biochemical and Biophysical Research Communications 451(3): 408-414.
Choi, J., S. R. Fauce and R. B. Effros (2008). "Reduced telomerase activity in human T lymphocytes exposed to cortisol." Brain Behav Immun 22(4): 600-605.
Epel, E. S., J. Lin, F. S. Dhabhar, O. M. Wolkowitz, E. Puterman, L. Karan and E. H. Blackburn (2010). "Dynamics of telomerase activity in response to acute psychological stress." Brain Behav Immun 24(4): 531-539.
Neugart, S., S. Baldermann, B. Ngwene, J. Wesonga and M. Schreiner (2017). "Indigenous leafy vegetables of Eastern Africa — A source of extraordinary secondary plant metabolites." Food Research International 100: 411-422.
Odongo, G. A., N. Schlotz, S. Baldermann, S. Neugart, B. Ngwene, M. Schreiner and E. Lamy (2018). "Effects of Amaranthus cruentus L. on aflatoxin B1- and oxidative stress-induced DNA damage in human liver (HepG2) cells." Food Bioscience 26: 42-48.
Schlotz, N., G. A. Odongo, C. Herz, H. Wassmer, C. Kuhn and F. S. Hanschen (2018). "Are Raw Brassica Vegetables Healthier Than Cooked Ones? A Randomized, Controlled Crossover Intervention Trial on the Health-Promoting Potential of Ethiopian Kale." 10(11).
Reviewer 5 Report
This manuscript evaluates the impact of 5 days of Ethiopian kale feeding (raw or cooked) on T cells, specifically telomerase activity in CD4+ cells and CD8+ cells. They report that CD4+ activity was unaffected by exposure to the kale, whereas CD8+ activity showed an increase in individuals after they ate cooked kale for 5 days. This single measure asks many questions, without giving answers. How would an increased CD8+ telomerase support health? Was 5 days needed, or was the effect a response to a single meal given 2 h prior to measurement? Although you report that polyphenolic content was not significantly altered by cooking, you refer to an earlier paper where complex phenolics broke down upon cooking to generate hundreds of percent increase in simple kaempferol glycosides – but here you do not measure this – can kaempferol glycosides have this effect in in vitro studies? Your conclusion is that you cannot determine if this is a measure of improved health. What would it take to determine this – age-stressed subjects or otherwise stressed subjects? in vitro studies of kaempferol?
This manuscript has very little data, which alone have little or no ready conclusions about what caused this, whether acute or chronic, what this change means for health. Given the questions generated, with almost no possible interpretations of the work at this time, it may be too early to publish.
minor points:
line 103 remove ‘again’ – it sounds as though you freeze-dried before cooking, then again after cooking.
line 168 ….which paralleled an increase (remove the ‘to’)
line 226 …regarded as a promising (add the word ‘as’)
Author Response
This manuscript evaluates the impact of 5 days of Ethiopian kale feeding (raw or cooked) on T cells, specifically telomerase activity in CD4+ cells and CD8+ cells. They report that CD4+ activity was unaffected by exposure to the kale, whereas CD8+ activity showed an increase in individuals after they ate cooked kale for 5 days. This single measure asks many questions, without giving answers.
How would an increased CD8+ telomerase support health?
Basically, higher endogenous telomerase activity has been associated with better immune function and health status (Boccardi, Esposito et al. 2013, Schutte and Malouff 2014). The constant expression of telomerase at low level in normal PBMC has been suggested to exert a function in proliferation of naïve and/or memory cells to maintain T cell homeostasis (O'Bryan, Woda et al. 2013). So, elevated telomerase activity in CD8+ T cells might help to maintain a healthy cytotoxic lymphocyte function and enhance their anti-viral ability. Using hTERT-transduced CD8+ T cells it has been suggested that maintenance of telomerase activity is capable to extend life span and preserve the cytotoxic properties of CD8+ cells (Rufer, Migliaccio et al. 2001). We explained this in the discussion part of the manuscript. p. 6, l. 209-216
Was 5 days needed, or was the effect a response to a single meal given 2 h prior to measurement?
So far published dietary intervention studies reporting on telomerase activity increase in PBMC were conducted for several weeks. However, it is known that telomerase activity is a dynamic factor that can quickly change. For example, in vitro mitogenic T cell stimulation or acute stress of volunteers has been shown to elevate telomerase levels in PBMC within hours or days (Epel, Lin et al. 2010); further, telomerase activity is known to change following a circadian rhythm (Chen, Wen et al. 2014). Thus, based on this knowledge we wanted to determine whether really long-term intervention with a diet rich in bioactive phytochemicals is necessary or if already a short-term intervention can lead to significant changes in the enzyme activity. The study provides for the first time evidence that telomerase activity in T cells can respond quickly (within less than a week) to a food intervention. Based on our study design (and considering a likely cumulative effect of continuous plant intake after a proper wash-out phase from the bioactive phytochemicals), the effect can only be interpreted as due to repeated food intervention.
Whether even shorter intervention periods (of even hours after a single intake) can already significantly modulate the enzyme cannot be said from current data.
Although you report that polyphenolic content was not significantly altered by cooking, you refer to an earlier paper where complex phenolics broke down upon cooking to generate hundreds of percent increase in simple kaempferol glycosides – but here you do not measure this – can kaempferol glycosides have this effect in in vitro studies?
In the manuscript we wrote that the total amount of polyphenolic compounds was comparable in the raw and cooked plant material (Schlotz, Odongo et al. 2018) and some of the polyphenols such as caffeoylquinic acid have also previously reported to be highly thermostable (Dawidowicz and Typek 2010). Further we stated that although several complex flavonoids were significantly degraded by the cooking process, less complex compounds, for instance, kaempferol glycosides, were enriched (Schlotz, Odongo et al. 2018). The chemical analysis of B. carinata was done as described in our previous paper (Schlotz, Odongo et al. 2018), we added the reference in the method and discussion sections. See discussion part on page 7, l. 235-240.
Kaempferol glycosides are common flavonoids that have a wide range of biological activity including anti-inflammatory, anti-oxidant, anti-microbial, immunomodulation, anti-cancer, neuroprotective effect, etc. (Calderon-Montano, Burgos-Moron et al. 2011). So far, for some polyphenols (including kaempferol) and ITC telomerase modulation in terms of inhibition has been shown in human cancer cells. This is regarded as a promising pharmacologic strategy to drive cancer cells into apoptosis (Sadava, Whitlock et al. 2007, Lamy, Herz et al. 2013, Herz, Hertrampf et al. 2014, Eitsuka, Nakagawa et al. 2018). However, the applied concentrations are usually over an order of magnitude higher than the plasma levels reached by dietary intervention. No in vitro study has been carried out by now to investigate the impact of these phytochemicals at physiologic concentrations on telomerase in normal cells of the immune system and thus it remains speculation which plant compound the effect could be attributed to. This can also be found in the discussion part (page 7, l. 240-247).
Your conclusion is that you cannot determine if this is a measure of improved health.
Our conclusion is that we cannot confirm a health promoting effect of the plant intervention simply based on this parameter. However, these data line up themselves with other observations by our group, which showed strong anti-genotoxic capacity (Schlotz, Odongo et al. 2018) as well as strong DNA repair capacity (Odongo, Skatchkov et al. 2019) in volunteers´ PBMC after cooked B. carinata intervention. The immune system strongly depends on clonal expansion and cell division and thus has evolved telomerase regulation as strategy for telomere maintenance in support of this need. In this context, we argued that the present findings could indeed support the theory of a beneficial effect. But we also stated that whether this modulation is really an indicator of improved health or rather a compensatory mechanism induced by adverse conditions still needs further clarification.
What would it take to determine this – age-stressed subjects or otherwise stressed subjects? in vitro studies of kaempferol? This manuscript has very little data, which alone have little or no ready conclusions about what caused this, whether acute or chronic, what this change means for health. Given the questions generated, with almost no possible interpretations of the work at this time, it may be too early to publish.
We are aware of the limitations of this study and thoroughly discussed them in the manuscript. However, we want to emphasize that this is the very first study to show that a short-term human vegetable intervention is able to increase telomerase levels in T cell subsets. Other studies on human volunteers need to further elucidate the consequences of this effect, investigate time kinetics and try to identify potential relevant small molecules. Further, we recommend that in future, it needs to be investigated whether a Brassica intervention has long-term effects on telomere extension in specific T cell subsets. We could confirm here, that allyl ITC, although a very potent bioactive molecule and made responsible for many observed health promoting effects, is not relevant for the effects reported here.
minor points:
line 103 remove ‘again’ – it sounds as though you freeze-dried before cooking, then again after cooking.
It has been removed.
line 168 ….which paralleled an increase (remove the ‘to’).
It has been corrected.
line 226 …regarded as a promising (add the word ‘as’).
It has been corrected.
Cited literature:
Boccardi, V., A. Esposito, M. R. Rizzo, R. Marfella, M. Barbieri and G. Paolisso (2013). "Mediterranean diet, telomere maintenance and health status among elderly." PLoS One 8(4): e62781.
Dawidowicz, A. L. and R. Typek (2010). "Thermal stability of 5-o-caffeoylquinic acid in aqueous solutions at different heating conditions." J Agric Food Chem 58(24): 12578-12584.
Eitsuka, T., K. Nakagawa, S. Kato, J. Ito, Y. Otoki, S. Takasu, N. Shimizu, T. Takahashi and T. Miyazawa (2018). "Modulation of Telomerase Activity in Cancer Cells by Dietary Compounds: A Review." Int J Mol Sci 19(2).
Herz, C., A. Hertrampf, S. Zimmermann, N. Stetter, M. Wagner, C. Kleinhans, M. Erlacher, J. Schuler, S. Platz, S. Rohn, V. Mersch-Sundermann and E. Lamy (2014). "The isothiocyanate erucin abrogates telomerase in hepatocellular carcinoma cells in vitro and in an orthotopic xenograft tumour model of HCC." J Cell Mol Med 18(12): 2393-2403.
Lamy, E., C. Herz, S. Lutz-Bonengel, A. Hertrampf, M. R. Marton and V. Mersch-Sundermann (2013). "The MAPK pathway signals telomerase modulation in response to isothiocyanate-induced DNA damage of human liver cancer cells." PLoS One 8(1): e53240.
O'Bryan, J. M., M. Woda, M. Co, A. Mathew and A. L. Rothman (2013). "Telomere length dynamics in human memory T cells specific for viruses causing acute or latent infections." Immun Ageing 10(1): 37.
Odongo, G. A., I. Skatchkov, C. Herz and E. Lamy (2019). "Optimization of the alkaline comet assay for easy repair capacity quantification of oxidative DNA damage in PBMC from human volunteers using aphidicolin block." DNA Repair (Amst) 77: 58-64.
Rufer, N., M. Migliaccio, J. Antonchuk, R. K. Humphries, E. Roosnek and P. M. Lansdorp (2001). "Transfer of the human telomerase reverse transcriptase (TERT) gene into T lymphocytes results in extension of replicative potential." Blood 98(3): 597-603.
Sadava, D., E. Whitlock and S. E. Kane (2007). "The green tea polyphenol, epigallocatechin-3-gallate inhibits telomerase and induces apoptosis in drug-resistant lung cancer cells." Biochemical and Biophysical Research Communications 360(1): 233-237.
Schlotz, N., G. A. Odongo, C. Herz, H. Wassmer, C. Kuhn and F. S. Hanschen (2018). "Are Raw Brassica Vegetables Healthier Than Cooked Ones? A Randomized, Controlled Crossover Intervention Trial on the Health-Promoting Potential of Ethiopian Kale." 10(11).
Schutte, N. S. and J. M. Malouff (2014). "A meta-analytic review of the effects of mindfulness meditation on telomerase activity." Psychoneuroendocrinology 42: 45-48.
Round 2
Reviewer 2 Report
I have no other comments. Thank you.
Reviewer 5 Report
The ms is well written, however the data are not readily interpreted for a number of reasons, Several are outlined by the authors such as the low number of subjects/ large variation in telomerase measurements or the lack of knowledge as to whether this is an acute or sub-chronic effect. Some reasons are left unspoken, such as the lack of a detailed analysis of kaempferol conjugates, in the intervention material or in plasma, leading to an acknowledged lack of knowledge of the active component. The authors correctly point out that this novel finding could lead to some interesting information if plant flavonoid(s) are able to impact telomerase. Yet there are so many unknowns, that one cannot come to any conclusions at this time.
line 202, 'affect' not 'effect.